# A Multicenter Cross-Sectional Survey of Knowledge, Attitude, and Practices of Healthcare Professionals towards Antimicrobial Stewardship in Ghana: Findings and Implications

**DOI:** 10.3390/antibiotics12101497

**Published:** 2023-09-29

**Authors:** Israel Abebrese Sefah, Sarentha Chetty, Peter Yamoah, Johanna C. Meyer, Audrey Chigome, Brian Godman, Varsha Bangalee

**Affiliations:** 1Pharmacy Practice Department, School of Pharmacy, University of Health and Allied Sciences, Volta Region, Ho PMB 31, Ghana; pyamoah@uhas.edu.gh; 2Department of Pharmacy and Pharmacology, Faculty of Health Sciences, University of the Witwatersrand, Johannesburg 2193, South Africa; sarentha.chetty@wits.ac.za; 3Department of Public Health Pharmacy and Management, School of Pharmacy, Sefako Makgatho Health Sciences University, Molotlegi Street, Garankuwa, Pretoria 0208, South Africa; hannelie.meyer@smu.ac.za (J.C.M.); audreykchigome@gmail.com (A.C.); 4South African Vaccination and Immunisation Centre, Sefako Makgatho Health Sciences University, Molotlegi Street, Garankuwa, Pretoria 0208, South Africa; 5Department of Pharmacoepidemiology, Strathclyde Institute of Pharmacy and Biomedical Science, University of Strathclyde, Glasgow G4 0RE, UK; 6Discipline of Pharmaceutical Sciences, School of Health Sciences, University of KwaZulu-Natal, Durban 4041, South Africa; bangalee@ukzn.ac.za

**Keywords:** antimicrobial resistance, antimicrobial stewardship programs, knowledge, attitude and practice, healthcare professionals, public hospitals, Ghana

## Abstract

Antimicrobial stewardship (AMS) programs are part of the key activities that contribute to reducing antimicrobial resistance (AMR). Good knowledge, attitudes, and practices (KAP) among healthcare professionals (HCPs) are essential to improving future antimicrobial use and reducing AMR, which is a priority in Ghana. A multicenter cross-sectional survey was conducted in six public hospitals in Ghana among key HCPs to assess their level of KAP towards AMS using a validated self-administered electronic questionnaire. Data analyses included descriptive and inferential statistics using STATA version 14. Overall, 339 out of 355 HCPs responded to the questionnaire, giving a response rate of 95.5%. Most responders were nurses (n = 256, 78.2%), followed by medical doctors (n = 45, 13.3%). The study recorded both poor knowledge (8.9%) and practice levels (35.4%), as well as a good attitude (78.8%) towards AMS. Ongoing exposure to AMS structured training, exposure to continuous professional development training on AMS in the previous year, and the number of years of working experience were predictors of the HCPs’ level of knowledge (aOR = 3.02 C.I = 1.12–8.11), attitude (aOR = 0.37 C.I = 0.20–0.69) and practice (aOR = 2.09 C.I =1.09–3.99), respectively. Consequently, concentrated efforts must be made to address current low levels of knowledge and poor practices regarding AMS among HCPs in Ghana as part of ongoing strategies in the National Action Plan to reduce AMR.

## 1. Introduction

There are concerns regarding the increase in antimicrobial resistance (AMR) globally, given its negative impact on morbidity, mortality, and costs [1,2,3,4,5]. As a result, AMR is increasingly being considered as the next pandemic if practical steps are not taken to address rising rates [6]. A key activity in reducing AMR across countries has been the development of the Global Action Plan (GAP) by the World Health Organization (WHO) in 2015 [7]. Countries have subsequently developed their own National Action Plans (NAPs); however, these are at different stages of development and implementation [8,9,10,11,12,13]. A major activity within the NAPs is a greater understanding of current antimicrobial utilization patterns in hospitals as well as ongoing initiatives to reduce AMR, which include the instigation of antimicrobial stewardship programs (ASPs). ASPs have been identified by the WHO as one of the three pillars of an integrated approach to health systems strengthening to improve future antimicrobial utilization, thereby reducing AMR. The other two pillars are the instigation of infection prevention and control (IPC) activities alongside improving medicine use and patient safety [14,15].

Antimicrobial stewardship (AMS) has been defined both as a coherent set of actions that promote the responsible use of antimicrobials and an organizational or system-wide healthcare strategy to promote the appropriate use of antimicrobials via the implementation of evidence-based interventions [14]. ASPs, using strategies that include education, prospective audits of antibiotic prescribing with feedback, antibiotic formulary restrictions, and enhancing guideline adherence, have been shown to reduce inappropriate antibiotic prescribing and any associated adverse effects alongside reducing healthcare costs whilst reducing AMR [16,17,18,19,20,21].

Ghana developed and implemented its NAP in 2017 to reduce AMR, which is being regularly monitored and updated [12,22,23,24]. However, there are concerns regarding its implementation, and notable among them is the availability of resources [25]. The need to optimize antimicrobial use via the effective implementation of ASPs in all health facilities in Ghana is an integral component of one of the five strategic objectives of Ghana’s NAP. This is important, given concerns generally with the level of AMR reported across sub-Saharan Africa, including Ghana [1,26,27,28,29,30]. In Ghana, AMR in hospitals is exacerbated by high levels of inappropriate prescribing. This includes the overuse of antibiotics from the WHO ‘Watch’ list of antibiotics with high levels of empiric prescribing [31,32,33], prolonged administration of antibiotics to prevent surgical site infections [20,34,35,36], as well as concerns with guideline adherence [37]. Contrary to this, other studies have suggested high adherence to guidelines in Ghana [36]. There has also been disquiet regarding available financial resources and personnel to fully undertake ASPs in low- and middle-income countries (LMICs) [38]. However, this is changing in Africa, with a number of ASPs being successfully introduced to improve future prescribing [20,39,40,41], with ongoing educational activities among African countries, including Ghana, to improve future prescribing [42,43,44].

To date, there have been few ASPs undertaken in Ghana compared with a number of other African countries [20,40]. Increasing the number of ASPs undertaken in Ghana will require, among other things, good knowledge, attitude, and practices (KAP) among key healthcare professionals (HCPs) towards ASPs. This is because studies have shown that HCPs’ KAP towards the principles and activities of AMS appreciably impacts the future appropriate use of antibiotics [45,46]. Alongside this, there have been concerns regarding variable knowledge and attitudes towards the use of antibiotics and ASPs among HCPs in African countries, including Ghana [47,48,49,50,51], which must be addressed. Furthermore, there have also been concerns about the level of knowledge of AMR and AMS among pharmacy, medical, nursing, and physician assistantship students in Ghana [52].

The levels of KAP regarding antibiotics among HCPs are commonly assessed using self-reported assessment questions [53,54,55,56]. A systematic review of HCPs’ KAP towards antibiotic prescribing in LMICs showed a good level of knowledge regarding antibiotics and high levels of knowledge and confidence regarding the prescribing of antibiotics [55]. However, while some studies have shown good KAP among HCPs toward antibiotic use, other studies have reported varied results [49,50,51,53,55,56,57,58,59,60]. In addition, some studies have shown good knowledge and attitude towards antibiotics and AMR but poor practices, whilst others have shown good knowledge and practices but poor attitudes towards AMS [57,58,59].

To date, there appear to be few studies that have assessed factors that influence HCPs’ KAP regarding AMS, especially in Ghana. One study conducted in Uganda showed an association between females and higher academic qualifications and good attitudes towards AMS [60]. There is also currently limited knowledge in Ghana on AMS activities; however, this is beginning to change with, as mentioned, educational activities being introduced in hospitals in Ghana regarding ASPs to improve knowledge and practice [42,44,61]. Alongside this, the development of an App to improve access and adherence to antibiotic guidelines in hospitals in Ghana has also been identified as part of the many solutions to address this gap [34,37,62].

Consequently, the objective of this study was to address this information gap by assessing HCPs’ current KAP regarding AMS in six hospitals in two (Greater Accra and Volta) regions in Ghana. We believe this is the first multicenter survey targeted towards assessing HCP’s KAP of AMS in Ghana. In view of this, the findings from this study could be useful in designing appropriate interventions to improve future antimicrobial use and curb rising AMR in Ghana.

## 2. Results

### 2.1. Participants Socio-Demographic and Other Characteristics

Out of the 355 participants who were expected to respond to the survey, 339 HCPs answered the questionnaire, giving a response rate of 95.5%. The majority of participants were female (n = 339, 75.5%), within the age group of 25 to 49 years (n = 292, 86.1%), followed by those below 25 years (n = 31, 9.1%), with a mean age of 33.16 ± 7.98 years. 

The majority of the HCP participants were from the Greater Accra Region (n = 242, 71.4%), were working in a primary care level health facility (n = 225, 66.4%), were from the nursing profession (n = 256, 78.2%), followed by medical doctors (n = 45, 13.3%) and pharmacists (n = 29, 8.6%), with diploma/higher national diploma (n = 256, 78.2%) educational qualification followed by bachelor’s degree (n = 128, 37.8%). Most participants had worked for between 10–19 years (n = 109, 32.2%), followed by 30 to 39 years (n = 100, 29.5%) (Table 1).

Most of the participants had no previous exposure to structured AMS training (n = 174, 51.3%) and no continuous professional development (CPD) training on AMS in the past year (n = 222, 65.5%).

### 2.2. Knowledge, Attitude, and Practice Scores

The majority of participants had poor knowledge (n = 309, 91.2%) and poor practice (n = 219, 64.6%) but a good attitude (n = 267, 78.8%) towards AMS (Table 2).

### 2.3. Factors Associated with Healthcare Professionals’ Knowledge, Attitude and Practice Scores

The Chi-square test showed a statistically significant association between the sex (*p* = 0.012) and the type of HCP (*p* = 0.023) as well as exposure to AMS structured training (*p* = 0.001), continuous professional development training on AMS in the previous year (*p* = 0.008) and HCPs’ level of knowledge of AMS as assessed with the questionnaire (Table 3).

The attitude level of the interviewed HCPs towards AMS also showed a statistically significant association with their sex (*p* = 0.005), the level of care of the health facility (*p* < 0.001), exposure to AMS structured training (*p* = 0.008) and exposure to continuous professional development (CPD) training on AMS in the previous year (*p* < 0.001) (Table 3).

HCPs’ practice of AMS showed a statistically significant association with the type of healthcare professionals (*p* = 0.010) and their years of professional experience (*p* = 0.035) (Table 3).

### 2.4. Predictors of Healthcare Professionals’ Level of Knowledge, Attitude, and Practice of AMS

HCPs’ level of knowledge was predicted using their exposure to AMS structured training. HCPs exposed to such training were three times more likely to have good knowledge of AMS compared to HCPs who had received no training (aOR = 3.02; 95% CI = 1.12–8.11) (Table 4).

The attitude level of HCPs was also predicted using the level of healthcare facility (aOR = 2.83; 95% CI =1.34–5.99) and their exposure to CPD training on AMS the previous year (aOR = 0.37; 95% CI = 0.20–0.69) (Table 4).

The practice level of HCPs was predicted using their age (aOR = 3.05; 95% =1.08–8.62), the type of HCP, i.e., medical doctor, nurse, and pharmacist (aOR = 0.41; 95% CI = 0.21–0.83) and the number of years of working experience (aOR = 2.09; 95%CI = 1.09–3.99) (Table 4).

### 2.5. Differences in Knowledge, Attitude, and Practice Levels among the Types of Healthcare Professionals

The Shapiro–Wilk test for normality showed that the attitude and practice scores of HCPs were normal, whereas the knowledge scores were not normal (Table 5). 

There was a statistically significant difference in the mean score of the attitude (*p* < 0.001) and practice (*p* < 0.001) of AMS among the HCPs according to the one-way ANOVA test. Post hoc test using Dunn’s pairwise test showed that pharmacists had better attitudes towards AMS than medical doctors (*p* = 0.010) and nurses (*p* < 0.001), and medical doctors had a better attitude than nurses (*p* = 0.004). Concerning the practices of AMS, medical doctors performed better than nurses (*p* < 0.001) (Table 6).

There was also a statistically significant difference in the median score of the knowledge of AMS (*p* = 0.009) among the HCPs. The Bonferroni post hoc test showed that pharmacists had better knowledge of AMS than nurses (*p* = 0.015) (Table 6).

## 3. Discussion

To the best of our knowledge, we believe this is the first study of its kind in Ghana. This is because our study was designed to compare the KAP of AMS principles and theories among three essential HCP groups in healthcare facilities across Ghana. The HCP groups included pharmacists, medical doctors, and nurses, and we used a multicenter approach with six public hospitals. Previous studies undertaken in LMICs have typically focused on one HCP group; alternatively, they were conducted in a single setting [50,51,53]. In addition, most of these studies focused on KAP of antibiotic use and AMR and not on the principles and strategies of AMS, which include clinician education, pre-authorization, and automatic stop orders [44,45,48,50,51,53,54,55].

As mentioned earlier, Ghana has made progress by introducing AMS programs in some hospitals as part of ongoing efforts to achieve the goals of NAP, together with the support of the Fleming Fund [42,44,61]. In addition, an App has been developed to improve access to antibiotic guidelines in hospitals [62]. However, a broad assessment of HCPs’ current KAP of AMS will add to the knowledge base with the aim of reducing AMR in Ghana.

Most (86.1%) of the HCPs were in the middle age (25 to 49 years) group, which is comparable to the findings in similar previous studies [57,58,59,60]. A greater proportion of HCPs in our study were nurses (78.2%) followed by medical doctors (13.3%) and pharmacists (8.6%), which corresponds with the proportions of 75.5% of respondents being female as most nursing practitioners in Ghana and other similar countries are usually female. 

Approximately three-quarters of our study participants were working in a primary-level facility; consequently, a greater portion of our KAP findings were from this sector. This is important as up to 90% or more of antibiotic used worldwide occurs in primary care facilities [63,64]. Over 50% of the HCPs surveyed had at least a bachelor’s degree as their highest level of education, with three-quarters of the practitioners having at least 10 years of working experience. Consequently, our study findings were not from newly employed staff nor those of lower educational background, which is similar to other studies [60,65]; however, at variance with others [57,59].

Despite their educational background and at least a decade of work experience, less than half of HCPs in our study had been exposed to structured training on AMS. In addition, only 34.5% of HCPs surveyed had undergone CPD training on AMS, which is a concern. Perhaps not surprisingly, overall, 91.2% of HCPs surveyed had poor knowledge of AMS. Their level of knowledge was associated with their sex (*p* = 0.012), type of HCP (either pharmacist, medical doctor, or nurse) (*p* = 0.023), exposure to structure training on AMS (*p* = 0.001), and whether they had received CPD training on AMS (*p* = 0.008). The level of knowledge of HCPs also differed according to the level of training. HCPs who had experienced structured training on AMS were three times more likely to have better knowledge compared with those with less experience (aOR = 3.02; CI = 1.12–8.11). This shows a direct association between the extent of training on AMS and the level of knowledge of HCPs.

Of equal concern was the differences in knowledge regarding AMS among the three different types of HCPs. Higher levels of knowledge were recorded among pharmacists compared to nurses (*p* = 0.015). These findings demonstrate the importance of introducing AMS into the curriculum for the training of all HCPs and opportunities for on-the-job training as part of efforts to ensure uniform multi-disciplinary knowledge as part of ongoing activities to curb increasing AMR rates in Ghana [44,61]. A similar recommendation has been seen in other African countries [56,66].

Encouragingly, approximately 8 out of 10 HCPs in our study had good attitudes towards AMS, similar to the findings in other developing countries [57,60]. The level of attitude towards AMS was associated with age (*p* = 0.005), level of care of the facility (*p* < 0.001), exposure to structured training on AMS (*p* = 0.008), and whether the HCP had received CPD training on AMS (*p* < 0.001). Our study found that the level of attitude was reduced by 63% (aOR = 0.37; 95% CI = 0.20–0.69) by those who undertook CPD activities the previous year. As a result, our findings appear inconsistent with other studies where continuous education of HCPs has shown a positive influence on their attitude towards antibiotic use and other healthcare practices [60,67,68]. We are not sure of the reasons for this. However, attitude levels were approximately three times higher among HCPs who worked in secondary levels of care compared to those working in primary care facilities. The difference could be due to the specialization of practices, e.g., infectious disease clinicians, antimicrobial pharmacists, and medical microbiologists, at secondary-level health facilities compared to primary care counterparts. As a result, HCPs in secondary care facilities are often trained with some specialized knowledge and skills, which will affect their attitudes towards AMS. 

Our study also found that while medical doctors had a better attitude towards AMS than nurses, pharmacists had a better attitude than both medical doctors (*p* = 0.010) and nurses (*p* < 0.001). Considering the importance of a good attitude, not only towards AMS but towards all areas of healthcare delivery, effective training programs must be designed that impact positively on attitudes among all HCPs [44]. This is because, despite better attitudes towards AMS, 64.6% of HCPs in our study were classified as having poor practices towards AMS due to poor levels of knowledge. The level of practice of AMS among HCPs was, however, associated with the type of HCP (*p* = 0.010) and their duration of experience (*p* = 0.035). Compared to medical doctors, nurses were found to have a 59% lower level of practice of AMS. There was, however, no difference in the level of practice between medical doctors and pharmacists and between pharmacists and nurses. The difference in the levels of practice among the HCPs could be due to the difference in their knowledge levels, as seen in this study. Encouragingly, HCPs with greater working experience appeared to have twice as likely higher practice levels (aOR = 2.09; 95% CI =1.09–3.99). This may have accounted for the differences seen between medical doctors and nurses. However, we cannot say this with certainty. In view of this, there needs to be continuous in-service orientation and training on AMS among all cadres of HCPs to ensure an improved uniform level of practice in the future in Ghana [61]. HCPs’ level of practice was also found to be about two times higher among those HCPs with increased years of experience (10–19 years) compared to those with lower years of experience (0–9 years).

We are aware that other studies have used lower than the ≥60% cut-off scores, while others have used higher scores up to ≥80%. However, we chose ≥60% based on previously published studies on this subject [52,56,67]. Consequently, comparisons of our findings with other similar studies must be performed with some circumspection [60,69,70]. In addition, the observed differences in the knowledge, attitude, and practice scores between the three groups of professionals were not adjusted further using multivariate analysis or by stratification, though our study identified their predictors. Our study was also limited by the exclusion of other key members of AMS hospital teams, including microbiologists and information technologists. However, physicians, pharmacists, and nurses are typically key members of AMS teams. Despite these limitations, we believe our findings are robust, providing future direction to key groups in Ghana.

## 4. Materials and Methods

### 4.1. Study Design

A hospital-based cross-sectional survey was conducted via an electronic self-administered structured questionnaire to assess the KAP of HCPs (medical doctors, pharmacists, and nurses) concerning ASP in six public hospitals in the Greater Accra and Volta Regions of Ghana.

### 4.2. Study Setting and Population

The study was conducted in 2 of the 16 administrative regions in Ghana, i.e., the Greater Accra Region, where the capital city of Ghana is located, and the Volta Region. Ghana’s population, according to the 2021 Population and Housing Census Report, currently stands at 30.8 million, with the Greater Accra Region being the second most populous region while the Volta Region is the seventh [71]. The two regions share a boundary in the southeastern part of the country, with the Greater Accra region being predominantly urban while the Volta region is predominantly rural. 

These two regions were purposively selected to represent the different dynamics in healthcare delivery among public Ghanaian hospitals, which are predominantly utilized by the citizens in Ghana due to their acceptance of the national insurance scheme to improve patient care [72,73].

The study was subsequently undertaken in six public health facilities within these two regions, with the hospitals conveniently sampled to ensure ease of data collection. Table 7 gives further details regarding the characteristics of these six facilities.

### 4.3. Data Collection

The KAP questionnaire was adapted from similar studies [49,61,65,74] and reviewed based on the considerable experience of the co-authors in this field. We have used a similar approach previously where no standard questionnaire existed [50,75,76].

The questionnaire was subsequently pre-tested among 31 HCPs from two hospitals in the two regions that were not part of the above-selected study sites to assess the clarity and reliability of the questions. The content validity ratio, calculated for all items to assess for relevance, ranged from 0.61 to 1.00, and that for clarity ranged from 0.65 to 1.00. The final scale-content validity index for the relevance and clarity of the questionnaire was 0.65 for both indices.

Suggestions made by the HCPs and experts were incorporated into the final questionnaire. The reliability analysis of the questionnaire was performed by estimating the Cronbach’s alpha value was 0.71. This showed that the final questionnaire had acceptable internal consistency.

The final questionnaire contained sections on the socio-demographic characteristics of participants as well as separate domain sections for assessing the KAP of AMS. Overall, the knowledge domain contained 10 questions with three (Yes, No, and Unsure) answer options for each question. A ‘wrong’ or an ‘unsure’ answer was given a zero score, and a correct answer was given a score of one. Total scores for each participant were subsequently converted to a percentage. The outcomes regarding knowledge were dichotomized as “good” versus “poor”, where a total score for knowledge ≥ 60% was considered good knowledge and <60% was considered poor knowledge.

The attitude and practice sections also had seven questions, each with Likert scale responses, which were scored using a 5-item rating scale (strongly agree = 5, agree = 4, neutral = 3, disagree = 2, and strongly disagree = 1) for each question, which is similar to other studies [77,78,79]. Outcomes regarding attitudes and practices were dichotomized as “good” versus “poor”, where a total score ≥ 60% was considered good and a total score < 60% was considered poor. The calculation of the scores for the KAP of HCPs was adapted from previous studies [56,67].

### 4.4. Sample Size and Sampling Methods

The calculated sample size for the number of HCPs (doctors, pharmacists, and nurses) using a margin of error of 5%, a 95% confidence level with a total population of 2798, a 50% assumed rate of appropriate KAP of ASP based on the findings from similar studies [49,61,65,74], and using the Raosoft online sample size calculator (http://www.raosoft.com/samplesize.html (accessed on 20 June 2023)) was 338. Using a 5% possible non-response rate, an upward adjustment of the sample size to 355 was made.

The required sample size for participants from each health facility was calculated using a proportional sampling method based on the total number of HCPs in the three categories of eligible staff as follows: Greater Accra Regional Hospital (140 out of 1111 HCPs), Tema General Hospital (70 out of 556 HCPs), LEKMA Hospital (68 out of 534 HCPs), Hohoe Municipal Hospital (35 out of 274 HCPs), Ho Municipal Hospital (20 out of 160 HCPs) and Keta Municipal Hospital (2). This was done to ensure a reasonable, proportional representation of the six health facilities and the three HCP groups.

The final questionnaire was subsequently distributed to a target of 355 HCPs working in the study sites using a web-based Google form (Appendix A) [49,61,65,74]. An online link to the questionnaire was shared with all the hospital clinical coordinators or administrators via email and WhatsApp for onward dissemination to their HCPs. Participants included in the KAP study comprised medical doctors, nurses, and pharmacists who were employed full-time. All other categories of HCPs were excluded.

### 4.5. Data Analysis

The data collected using the Google online form were imported into a Microsoft Excel sheet, cleaned, and thereafter exported to STATA version 14 for statistical analysis. Descriptive statistics, bivariate, i.e., the Chi-square test of independence, and multivariate, i.e., multiple logistic regression, tests were performed to determine the proportion of staff with good KAP of AMS and the factors associated with their occurrence.

The normality of the KAP scores of HCPs was assessed using the Shapiro–Wilk test. Differences in the mean and median scores of the KAP of AMS among the different types of HCPs were determined using the one-way analysis of variance test (ANOVA). The Kruskal–Wallis test, a non-parametric test, was used to determine the differences in the median scores of knowledge among the different HCPs. Post hoc tests were used to determine which of the types of HCPs had better KAP scores than others.

### 4.6. Ethical Considerations

The web-based google questionnaire was designed in order that consent had to be given online by participating HCPs before access to the questionnaire was granted. Detailed information about the survey was provided to the participants on the need for voluntary participation and withdrawal and how potential risks have been addressed. 

HCP data privacy and confidentiality were also ensured by anonymizing all identifiers of the participants, and electronic data collated were archived in a zip file and password protected. 

Ethical clearance was obtained from both the University of KwaZulu-Natal (BREC/00004236/2022) and the Ghana Health Service (GHS-ERC/01708/22), and administrative approval was also sought from the management of each of the six participating hospitals.

## 5. Conclusions and Next Steps

Overall, there was poor knowledge and practices of AMS among the HCPs surveyed in our study, though their attitude level was appreciable. There were also variable levels of KAP among medical doctors, nurses, and pharmacists. Various factors, including exposure to AMS structured training and CPD training on AMS the previous year, the type of HCP with their number of years of working experience, were predictors of the HCPs’ levels of KAP, respectively. 

Effective and sustainable efforts must be made by all key stakeholder groups to tackle the low levels of knowledge and skills for the practice of AMS among HCPs going forward. Proposed activities include reviewing the current curricula for student HCPs among Universities in Ghana as well as ongoing CPD activities post-qualification. These strategies are essential to reduce AMR rates in Ghana in line with the goals of the NAP. We will continue to monitor the situation.

## Figures and Tables

**Table 1 antibiotics-12-01497-t001:** Descriptive statistics of participants’ characteristics.

Variable (n = 339)	Frequency (%)
**Age (years)**	
≤24	31 (9.1)
25–49	292 (86.1)
≥50	16 (4.7)
**Sex**	
Male	83 (24.5)
Female	256 (75.5)
**Regional Location**	
Greater Accra	242 (71.4)
Volta	97 (28.6)
**Level of Care**	
Primary	225 (66.4)
Secondary	114 (33.6)
**Healthcare Professional**	
Pharmacist	29 (8.6)
Medical Doctor	45 (13.3)
Nurse	265 (78.2)
**Highest Level of Education**	
PhD/Fellowship	9 (2.7)
Masters/Membership	45 (13.3)
Bachelor Degree	128 (37.8)
Diploma/Higher National Diploma	151 (44.5)
Certificate	6 (1.8)
**Experience (years)**	
0–9	82 (24.2)
10–19	109 (32.2)
20–29	48 (14.2)
30–39	100 (29.5)
**Exposure to AMS structured training**	
Yes	165 (48.7)
No	174 (51.3)
**Continuous Professional Development Training on AMS in the last year**	
Yes	117 (34.5)
No	222 (65.5)

**Table 2 antibiotics-12-01497-t002:** Knowledge, attitude, and practice scores.

Variable (n = 339)	Frequency (%)
**Knowledge Level**	
Good	30 (8.9)
Poor	309 (91.2)
**Attitude Level**	
Good	267 (78.8)
Poor	72 (21.2)
**Practice Level**	
Good	120 (35.4)
Poor	219 (64.6)

**Table 3 antibiotics-12-01497-t003:** Factors associated with healthcare professionals’ knowledge, attitude, and practice levels.

Variable	Knowledge Level			Attitude Level			Practice Level		
	Good n (%)	Poor n (%)	*p*-Value	Good n (%)	Poor n (%)	*p*-Value	Good n (%)	Poor n (%)	*p*-Value
Age (years) (n = 339)			0.464			0.005 *			0.055
≤24	1 (3.2)	30 (96.7)		18 (58.1)	13 (41.9)		5 (16.1)	26 (83.8)	
25–49	27 (9.2)	265 (90.8)		234 (80.1)	58 (19.8)		110 (37.6)	182 (62.3)	
≥50	2 (12.5)	14 (87.5)		15 (93.7)	1 (6.3)		5 (31.2)	11 (68.7)	
Sex (n = 339)			0.012 *			0.615			0.669
Male	13 (15.7)	70 (84.3)		67 (80.7)	16 (19.3)		31 (37.3)	52 (62.7)	
Female	17 (6.6)	239 (93.4)		200 (78.2)	56 (21.8)		89 (34.7)	167 (65.3)	
Regional Location (n = 339)			0.860			0.100			0.503
Greater Accra	21 (8.7)	221 (91.3)		185 (76.5)	57 (23.5)		83 (34.3)	159 (65.7)	
Volta	9 (9.3)	88 (90.7)		82 (84.5)	15 (15.5)		37 (38.2)	60 (61.8)	
Level of Care (n = 339)			0.660			<0.001 *			0.932
Primary	21 (9.3)	204 (90.7)		163 (72.5)	62 (27.5)		80 (35.6)	145 (64.4)	
Secondary	9 (7.9)	105 (92.1)		104 (91.2)	10 (8.8)		40 (35.1)	74 (64.9)	
Healthcare Professional (n = 339)			0.023 *			0.093			0.010 *
Pharmacist	6 (20.6)	23 (79.3)		26 (89.6)	3 (10.4)		10 (34.5)	19 (65.5)	
Medical Doctor	6 (13.3)	39 (86.7)		39 (86.7)	6 (13.3)		25 (55.6)	20 (44.4)	
Nurse	18 (6.7)	247 (93.2)		202 (76.2)	63 (23.8)		85 (32.1)	180 (67.9)	
Highest Level of Education (n = 339)			0.114						0.058
PhD/Fellowship	3 (33.3)	6 (66.7)		7 (77.8)	2 (22.3)		6 (66.7)	3 (33.3)	
Masters/Membership	4 (8.9)	41 (91.1)		40 (88.9)	5 (11.1)		22 (48.9)	23(51.1)	
Bachelor Degree	10 (7.8)	118 (92.2)		120 (93.7)	8 (6.3)		44 (34.4)	84 (65.6)	
Diploma/Higher National Diploma	13 (8.6)	138 (91.4)		97 (64.2)	54 (35.8)		46 (30.5)	105 (69.5)	
Certificate	0 (0.0)	6 (100.0)		3 (50.0)	3 (50.0)		2 (33.3)	4 (66.7)	
Experience (years) (n = 339)			0.506			0.223			0.035 *
0–9	6 (7.3)	76 (92.6)		59 (72.0)	23 (28.0)		21 (25.6)	61 (74.4)	
10–19	9 (8.2)	100 (91.8)		92 (84.4)	17 (15.6)		49 (44.9)	60 (55.0)	
20–29	7 (14.6)	41 (85.4)		38 (79.2)	10 (20.8)		14 (29.2)	34 (70.8)	
30–39	8 (8.0)	92 (92.0)		78 (78.0)	22 (22.0)		36 (36.0)	64 (64.0)	
Exposure to AMS structured training (n = 339)			0.001 *			0.008 *			0.297
Yes	23 (13.9)	142 (86.1)		120 (72.7)	45 (27.3)		63 (38.2)	102 (61.8)	
No	7 (4.1)	167 (95.9)		147 (84.5)	27 (15.5)		57 (32.7)	117 (67.2)	
Continuous Professional Development Training on AMS last year (n = 339)			0.008 *			<0.001 *			0.392
Yes	17 (14.5)	100 (85.5)		76 (65.0)	41 (35.0)		45 (38.5)	72 (61.5)	
No	13 (5.9)	209 (94.1)		191 (86.0)	31 (14.0)		75 (33.8)	147 (66.2)	

NB: Independent variables with *p*-values asterisked showed statistically significant association with the outcome variables (knowledge, attitude, and practice levels).

**Table 4 antibiotics-12-01497-t004:** Multiple logistic regression between independent variables and study outcomes.

Variable	Knowledge Level		Attitude Level		Practice Level	
	Adjusted Odds Ratio	95% CI	Adjusted Odds Ratio	95% CI	Adjusted Odds Ratio	95% CI
Age (years) (n = 339)						
≤24 (r)	1.00		1.00		1.00	
25–49	2.69	0.33–22.18	1.65	0.70–3.89	3.05	1.08–8.62 ^#^
≥50	3.19	0.21–49.08	4.76	0.47–48.28	3.13	0.63–15.67
Sex (n = 339)						
Male	1.57	0.63–3.92	1.01	0.48–2.14	0.91	0.49–1.64
Female (r)	1.00		1.00		1.00	
Level of Care (n = 339)						
Primary (r)	1.00		1.00		1.00	
Secondary	0.76	0.31–1.85	2.83	1.34–5.99 ^#^	0.84	0.50–1.39
Healthcare Professional (n = 339)						
Pharmacist	1.24	0.33–4.69	1.80	0.39–8.32	0.39	0.15–1.07
Medical Doctor (r)	1.00		1.00		1.00	
Nurse	0.45	0.15–1.38	0.68	0.25–1.87	0.41	0.21–0.83 ^#^
Experience (years) (n = 339)						
0–9 (r)	1.00		1.00		1.00	
10–19	1.17	0.37–3.69	1.89	0.87–4.09	2.09	1.09–3.99 ^#^
20–29	2.63	0.73–9.49	1.19	0.45–3.11	1.13	0.47–2.72
30–39	1.11	0.34–3.56	1.12	0.53–2.36	1.49	0.77–2.91
Exposure to AMS structured training (n = 339)						
Yes	3.02	1.12–8.11 ^#^	0.76	0.40–1.44	1.26	0.75–2.12
No (r)	1.00		1.00		1.00	
Continuous Professional Development Training on AMS last year (n = 339)						
Yes	1.70	0.71–4.06	0.37	0.20–0.69 ^#^	1.25	0.72–2.15
No (r)	1.00		1.00		1.00	

NB: Confidence intervals (CI) with the hashtag (^#^) symbol are statistically significant.

**Table 5 antibiotics-12-01497-t005:** Shapiro–Wilk W test for normality of outcome variables.

Variable	Observation	*p*-Value	Interpretation
Knowledge score	339	<0.001	Not normal
Attitude score	339	0.163	Normal
Practice score	339	0.306	Normal

**Table 6 antibiotics-12-01497-t006:** Comparison of the knowledge, attitude, and practice scores by the type of healthcare professional.

Variable	Test Statistics	*p*-Value	Post hoc Test Results
Knowledge score	X^2^ = 9.34	0.009	* Pharmacist > Nurse (*p* = 0.015)
			* Medical > Nurse (*p* = 0.061)
			* Pharmacist > Medical doctor (*p* = 0.703)
Attitude score	F = 22.69	<0.001	^#^ Medical doctor > Nurse (*p* = 0.004)
			^#^ Pharmacist > Medical doctor (*p* = 0.010)
			^#^ Pharmacist > Nurse (*p* < 0.001)
Practice score	F = 7.88	<0.001	^#^ Medical doctor > Nurse (*p* < 0.001)
			^#^ Medical doctor > Pharmacist (*p* = 0.509)
			^#^ Pharmacist > Nurse (*p* = 0.416)

NB: * The Dunn’s pairwise test compares knowledge scores by healthcare professionals after the Kruskal–Wallis Rank test; ^#^ The Bonferroni test compares attitude scores by healthcare professionals after one-way analysis of variance test between attitude score and the type of healthcare professional; *p*-values for the post hoc test highlighted in bold showed statistical significance.

**Table 7 antibiotics-12-01497-t007:** Characteristics of the six participating hospitals.

Region	Hospital	Characteristics Including Number of Beds
**Greater Accra**	Greater Accra Regional Hospital	Secondary-level public hospital with a 600-bed capacity Serves as a referral site for all primary care facilities in the region as the hospital provides general and specialized care services to patients
	Tema General Hospital	Located in the Team Metropolitan Area A 400-bed capacity facility providing primary healthcare-level services, including internal medicine, general surgery, pediatrics, theatre, obstetrics, gynecological, accident, and emergency services Additionally, providing some specialized care including ophthalmology, dental, diabetic, sickle cell and dermatology services
	Ledzokuku-Krowor Municipal Assembly Hospital	Popularly called LEKMA Hospital 100-bed capacity primary-level facility located at Teshie in the Greater Accra region LEKMA hospital currently provides general services, including internal medicine, general surgery, pediatrics, theatre, obstetrics, gynecological, and accident and emergency services, as well as pharmaceutical care
**Volta**	Hohoe Municipal Hospital	178-bed capacity facility with 467 staff Serves as a secondary-level health facility in the region Major referral site for health facilities in the Volta and Oti Regions as it provides services ranging from internal medicine, general surgery, pediatrics, theatre, obstetrics, gynecological, and accident and emergency services to laboratory services and pharmaceutical care
	Keta Municipal Hospital	Located in the Keta Municipality Has a staff strength of 273, comprising medical doctors, pharmacists, general nurses, midwives, allied health staff, and other support staff Provides general services including internal medicine, general surgery, pediatrics, theatre, obstetrics, gynecological, accident and emergency service and specialized care including ophthalmology, dental and diabetic care
	Ho Municipal Hospital	Located in Ho, the capital city of the Volta region 150-bed capacity facility with approximately 150 nurses Provides general and specialized outpatient care and inpatient care to patients that utilize its services

## Data Availability

Additional data is available from the corresponding authors on reasonable request.

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
