# Peer review of "A Multicenter Cross-Sectional Survey of Knowledge, Attitude, and Practices of Healthcare Professionals towards Antimicrobial Stewardship in Ghana: Findings and Implications"

_antibiotics, 2023, doi:10.3390/antibiotics12101497_

Round 1

Reviewer 1 Report

Overall a very pertinent and through study. The selection of institutions and region are very relevant. A possible reason for difference in the knowledge , attitude and practice between the three group of professionals may be included in the discussion portion.

Author Response

Quality of English Language

( ) I am not qualified to assess the quality of English in this paper
( ) English very difficult to understand/incomprehensible
( ) Extensive editing of English language required
( ) Moderate editing of English language required
( ) Minor editing of English language required
(x) English language fine. No issues detected

Author comments: Thank you – appreciated!

Yes

Can be improved

Must be improved

Not applicable

Does the introduction provide sufficient background and include all relevant references?

(x)

( )

( )

( )

Are all the cited references relevant to the research?

(x)

( )

( )

( )

Is the research design appropriate?

(x)

( )

( )

( )

Are the methods adequately described?

(x)

( )

( )

( )

Are the results clearly presented?

(x)

( )

( )

( )

Are the conclusions supported by the results?

(x)

( )

( )

( )

Comments and Suggestions for Authors

1) Overall a very pertinent and through study. The selection of institutions and region are very relevant.

Author comments: Thank you for these kind words – appreciated!

2) A possible reason for difference in the knowledge, attitude and practice between the three group of professionals may be included in the discussion portion.

Author comments: Thank you for your comment. We have stated in our manuscript that the main predictor of attitude was the level of care possibly due to specialization of practices seen among those in secondary level of care. Our study found that the difference in the level of practice was predicted by the number of years of working experience between medical doctors and nurses. We have also stated that the difference in the level of knowledge among them was predicted by their exposure to structured training on AMS - which may have been due to the differences in curriculum for the training of these professionals. However, we cannot say this with certainty.

We have now added to our limitations that the differences observed among them could due to confounders since the observed differences were not adjusted by a further multivariate analysis using the significant variables or by stratification. We trust this is now acceptable.

Reviewer 2 Report

This is a well-designed and conducted study that brings innovation in its field.

1) Is the sample proportional to the population? Or were there differential lost in some place? Please inform this in the Results topic.

2) Is the position of "Materials and Methods" correct?

3) I strongly recommend a stratified representation of results in Table 6, because of confusion factors in professional competence, in variables such as:

- Level of Care

- Experience (years)

- Exposure to AMS structured training

- Continuous Professional Development Training on AMS last year

Author Response

Open Review

Quality of English Language

( ) I am not qualified to assess the quality of English in this paper
( ) English very difficult to understand/incomprehensible
( ) Extensive editing of English language required
( ) Moderate editing of English language required
( ) Minor editing of English language required
(x) English language fine. No issues detected

Author comment: Thank you – appreciated!

Yes

Can be improved

Must be improved

Not applicable

Does the introduction provide sufficient background and include all relevant references?

(x)

( )

( )

( )

Are all the cited references relevant to the research?

(x)

( )

( )

( )

Is the research design appropriate?

(x)

( )

( )

( )

Are the methods adequately described?

(x)

( )

( )

( )

Are the results clearly presented?

( )

(x)

( )

( )

Are the conclusions supported by the results?

(x)

( )

( )

( )

Comments and Suggestions for Authors

1) This is a well-designed and conducted study that brings innovation in its field.

Author comments: Thank you for these kind words – appreciated

2) Is the sample proportional to the population? Or were there differential lost in some place? Please inform this in the Results topic.

Author comments: Thank you for this. The sampling of the various professionals in the difference health facilities were done using a calculated proportional sampling method which has been stated in the method section under the sampling method heading. We hope this explanation is helpful.

3) Is the position of "Materials and Methods" correct?

Author comments: Thank you – yes this is the order in the Template supplied by Antibiotics. I trust this is OK with you

4) I strongly recommend a stratified representation of results in Table 6, because of confusion factors in professional competence, in variables such as:

- Level of Care

- Experience (years)

- Exposure to AMS structured training

- Continuous Professional Development Training on AMS last year

Author comments: Thank you for this comment. We have now added to our limitations that the difference in the level of knowledge, attitude and practice observed among the participants could due to confounders since the observed differences were not adjusted for by multivariate analysis using the significant variables or by stratification. We trust this is now acceptable.